# Comparative Study of Vaccinated and Unvaccinated Hospitalised Patients: A Retrospective Population Study of 500 Hospitalised Patients with SARS-CoV-2 Infection in a Spanish Population of 220,000 Inhabitants

**DOI:** 10.3390/v14102284

**Published:** 2022-10-17

**Authors:** José M. Ruiz-Giardin, Marta Rivilla, Nieves Mesa, Alejandro Morales, Luis Rivas, Aída Izquierdo, Almudena Escribá, Juan V. San Martín, David Bernal-Bello, Elena Madroñal, Ana I. Farfán, Marta Guerrero, Ruth Calderón, Miguel A. Duarte, Sara I. Piedrabuena, María Toledano-Macías, José Á. Satué, Jorge Marrero, Cristina L. de Ancos, Begoña Frutos, Rafael Cristóbal, Guillermo Soria, Ibone Ayala-Larrañaga, Lorena Carpintero, Miguel de Hita, Celia Lara, Álvaro R. Llerena, Virginia García, Raquel Jiménez, Vanesa García, Elena M. Saiz-Lou, Santiago Prieto, Natalia González-Pereira, Luis Antonio Lechuga, Jorge Tarancón, Sonia Gonzalo

**Affiliations:** 1Medicina Interna, Hospital Universitario de Fuenlabrada, 28942 Madrid, Spain; 2CIBERINFEC, 28029 Madrid, Spain; 3Oficina Regional de Coordinación de Transplante de la Comunidad de Madrid, 28046 Madrid, Spain; 4Laboratorio Clínico, Hospital Universitario de Fuenlabrada, 28942 Madrid, Spain; 5Sistemas, Hospital Universitario de Fuenlabrada, 28942 Madrid, Spain

**Keywords:** COVID-19, SARS-CoV-2, vaccination, hospital admission, ICU admission, death

## Abstract

Objectives. This study aimed to compare the characteristics of fully and partially vaccinated or unvaccinated coronavirus disease 2019 (COVID-19) patients who were hospitalised in a population of 220,000 habitants. Methods: Retrospective, observational, and population studies were conducted on patients who were hospitalised due to COVID-19 from March to October 2021. We assessed the impact of vaccination and other risk factors through Cox multivariate analysis. Results: A total of 500 patients were hospitalised, among whom 77 (15.4%) were fully vaccinated, 86 (17.2%) were partially vaccinated, and 337 (67.4%) were unvaccinated. Fully vaccinated (FV) patients were older and had a higher Charlson index than those of partially vaccinated and unvaccinated patients (NFV). Bilateral pneumonia was more frequent among NFV (259/376 (68.9%)) than among FV patients (32/75 (42.7%)). The former had more intensive care unit admissions (63/423) than the latter (4/77); OR: 2.80; CI (1.07–9.47). Increasing age HZ: 1.1 (1.06–1.14)) and haematological disease at admission HZ: 2.99 (1.26–7.11)) were independent risk factors for higher mortality during the first 30 days of hospitalisation. The probability of an earlier discharge in the subgroup of 440 patients who did not die during the first 30 days of hospitalisation was related to age (older to younger: HZ: 0.98 (0.97–0.99)) and vaccination status. Conclusions: Among the patients hospitalised because of COVID-19, complete vaccination was associated with less severe forms of COVID-19, with an earlier discharge date. Age and haematological disease were related to a higher mortality rate during the first 30 days of hospitalisation.

## 1. Introduction

Two and a half years after the onset of the coronavirus disease 2019 (COVID-19) pandemic, more than 554 million cases and 6.3 million deaths have been reported worldwide. In Spain, 12.9 million cases and 108,000 deaths have been confirmed [1], of which an estimated 14.3% is registered from the Community of Madrid [2,3].

The Hospital Universitario de Fuenlabrada (HUF) is a second-level hospital that serves a population of approximately 220,000 people in the south of Madrid (of which 10% are not Caucasian [4] based on their place of birth) as it is currently the referral hospital for the admission of patients diagnosed with COVID-19 in this area.

Massive vaccination campaigns have become the main strategy against the COVID-19 pandemic. Messenger RNA (mRNA) vaccines (BNT162b2 and mRNA-1273) were the first to demonstrate protection against severe cases [5,6]. Later, vaccines based on defective viral vectors (ChAdOx1 nCoV-19/AZD1222 (AstraZeneca/Oxford)) showed an efficacy of 74% [7] and the Ad26.COV2.S single-dose vaccine of 66.9% in preventing COVID-19. These efficacy rates in preventing severe forms of the disease increased to up to 76.7% and 85.4% at days 14 and 28 after vaccination, respectively [8].

In Spain, the vaccination program began on 27 December 2020 [9] after the aforementioned vaccines were approved by the European Medicines Agency [10].

According to official data from the Spanish Ministry of Health by 30 September 2021, a total of 75.2% of the population of the Community of Madrid had completed the vaccination schedule, while 78.2% had received at least one dose of the vaccine. These percentages were very similar to those observed overall in Spain (77.3% and 79.5%, respectively) [11].

A study conducted in Israel [12] found that vaccination against severe acute respiratory syndrome coronavirus-2 (SARS-CoV-2) was associated with a significant reduction in infections, hospital admissions, cases of severe disease, and death related to COVID-19, especially in patients aged 65 years or older and among those who completed the two-dose schedule.

Few studies to date have described how the evolution is related to the vaccination of patients that need hospitalisation.

The present study aimed to describe patients hospitalised due to COVID-19 and detail the characteristics of vaccinated and unvaccinated patients and those related to hospital discharge and death.

## 2. Materials and Methods

### 2.1. Study Design

This retrospective cohort study included all patients aged 16 years or older who were admitted to HUF due to confirmed acute SARS-CoV-2 infection (defined as a suggestive clinical presentation and a positive detection of SARS-CoV-2 RNA using the nucleic acid amplification test mainly by TMA (gen N or gen ORF1ab) or reverse transcription polymerase chain reaction (various platforms with the detection of gen S or N1 + N2 or S or N + E) or antigen test (Abbott panbio) mainly from nasopharyngeal swab This retrospective cohort study included all patients aged 16 years or older admitted in the HUF due to confirmed acute SARS-CoV-2 infection (defined as a suggestive clinical presentation and a positive detection of SARS-CoV2 RNA by NAAT (nucleic acid amplification test) mainly by TMA (gen N or gen ORF1ab) or RT-PCR (various platforms with the detection of gen S or N1 + N2 or S or N + E) or antigen test (Abbott panbio) mainly from nasopharyngeal swab samples from 2 March to 27 September 2021.

Patients were considered fully vaccinated if they had received the second dose of either BNT162b2 mRNA, mRNA-1273, or ChAdOx1 nCoV-19/AZD1222 vaccines, or a single dose of the Ad26.COV2.S vaccine, and if the last dose had been administered at least 14 days before symptom onset (this group is hereafter referred to as “vaccinated” (FV)). The patients who did not fulfil both conditions were classified as not fully vaccinated (“not vaccinated” (NFV)).

Baseline epidemiological, clinical, laboratory, and radiologic data were collected in all cases as well as information regarding clinical evolution and COVID-19 treatment during hospitalisation (Appendix A).

At the end of their hospitalisation, the patients were classified depending on the severity of the disease: mild (without pneumonia), moderate (pneumonia without severe symptoms), or severe (pneumonia with severe symptoms (tachypnea or respiratory frequency over 22 rpm, saturation under 93%, or more than 50% of the lungs were affected, as seen in radiographic films)).

### 2.2. Data Collection

In 2020, the Clinical Investigation Ethics Committee of the HUF approved the project named “Database of COVID-19 patients treated in the Hospital Universitario de Fuenlabrada (FUENCOVID),” whose main objective was to create a large-scale database with anonymised and non-traceable medical information of patients with COVID-19 who were admitted to the HUF. All of the data in the present study were from FUENCOVID.

All of the patients in the study underwent a blood test upon admission (including among other parameters, C-reactive protein, D-dimer, lactate dehydrogenase (LDH), and ferritin levels, and lymphocyte count). The analytical parameters were monitored every 24–48 h according to the clinical condition of the patient. Upon admission, the patients’ chest radiographs were also recorded. Treatments were used by clinicians following the HUF protocol of treatment.

### 2.3. Statistical Analysis

Qualitative variables were described using the absolute value (N) and percentage (%). The comparison between qualitative variables was carried out using the Chi-square or Fisher’s exact/Yates’ tests (if more than 20% of less than five of the expected frequencies were observed).

For the quantitative variables, the normality of the distribution was checked using the Kolmogorov–Smirnov contrast and graphic tests. Variables with normal distribution were described using the mean ± standard deviation and others by the median plus interquartile range (IQR). Normal variables were compared using the mean comparison test for two independent samples with a previous Levene’s test (homogeneity of variances) and non-normal variables using the non-parametric Mann–Whitney U test.

We used a multivariable Cox proportional hazards regression analysis, which was adjusted for other confounding variables, to compare the evolution of FV and NFV patients during the first 30 days of hospital admission.

Two regression models were considered for 483 cases after excluding 17 cases that had unknown discharge dates: in the first analysis, the risk of death was compared between vaccinated and unvaccinated patients in the first 30 days of hospital admission (27 deaths were recorded in this period), and the second compared the risk of being discharged during the first 30 days of hospital admission in the subpopulation of 440 patients who did not die during that period. In the multivariate model, all of the demographic variables and comorbidities were included as possible confounding variables with *p* < 0.10 in the univariate Cox analysis.

The proportional hazards assumption was checked using a scaled Shoenfeld residual using both hypothesis testing and graphical methods. The linearity assumptions were checked plotting the Martingale residuals against continuous covariates.

Statistical analyses were performed using SPSS 25.0 software (IBM International Business Machines Corporation, New York, NY, USA) and R 4.2.1. software.

## 3. Results

A total of 3854 SARS-CoV-2 infected patients were admitted to the hospital between March 2020 and October 2021. The median age of inpatients and both intensive care unit (ICU) admissions and deaths had progressively decreased throughout the COVID-19 pandemic (Table 1) (Figure 1).

In the present study, 500 patients who were admitted from 21 March to 27 September 2021 (fourth and fifth pandemic waves, which coincided with the implementation of the vaccination program in Spain) were analysed. Among them, 77 patients (15.4%) were considered FV, and were most commonly vaccinated (79.2%) with the BNT162b2 mRNA vaccine, while 423 (84.6%) were included in the NFV group, among which 337 had not received any vaccine dose, and 86 had only received the first dose or had been infected with SARS-CoV-2 during the first 14 days after the last dose (Appendix A).

### 3.1. Age and Comorbidities

The FV patients’ median age (76 years) was significantly higher (*p* value = 0.001) than that of the NFV patients (55 years). The relative frequencies of patients with hypertension, diabetes mellitus, and dementia as well as of those who were nursing home residents were all significantly higher (*p* = 0.001) in the FV group compared to those in the NFV group. The proportion of haematologic and oncologic diseases, smokers, and chronic obstructive pulmonary disease were also higher in the FV group; however, these differences were not significant. Nevertheless, FV patients had a significantly greater Charlson comorbidity index than those of the NFV patients (2 versus 0) (Table 2).

### 3.2. Chest Radiograph upon Admission

Bilateral pneumonia at admission was more frequent among the NFV patients (259 (68.9%) than those FV patients 32 (42.7%); odds ratio (OR), 2.97; 95% confidence interval CI (1.79–4.93) *p* = 0.001) (Table 3).

### 3.3. Oxygen Supplementation

The NFV patients needed oxygen delivery both through Venturi (31%, 35%, and 50% oxygen) and non-rebreather masks more frequently than the FV patients; however, these differences were not significant (Table 3).

### 3.4. Laboratory Results

Median peak ferritin levels were significantly higher in the NFV group (700 ng/mL (IQR, 1268–336) versus 351 ng/mL (IQR, 739–144); *p* = 0.001). Although the median peak LDH, D-dimer and pre-tocilizumab interleukin-6 levels were all higher among the NFV patients than those FV patients, these differences were not significant. Likewise, the median nadir lymphocyte count during hospitalisation was lower in the NFV group (875 cells/μL than those FV 930 cells/μL), although the difference was non-significant (Table 3).

### 3.5. Treatments

Although there were no differences regarding corticosteroid and low-weight heparin administration among both groups, more patients in the NFV group received tocilizumab (202 out of 423 (47.8%) than those of FV 26/77 (33.8%); OR, 1.64; 95% CI (1.06–2.54); *p* = 0.025) (Table 4).

### 3.6. COVID-19 Severity: ICU Admission, Classification at Hospital Discharge, and In-Hospital Mortality

In total, 63 patients in the NFV group required ICU admission, compared to only four patients in the FV group (OR, 2.80; 95% CI (1.07–9.47)) (Table 5). A total of 17 patients (nine and eight from the NFV and FV groups, respectively) who died in the hospital were not previously considered eligible for ICU admission.

At the time of hospital discharge, he COVID-19 severity was assessed. We found that severe COVID-19 was more frequent among the NFV patients (209 (50.4%) than among FV 28 (37.3%); OR, 1.70; 95% CI (1.02–2.82); *p* = 0.001) (Table 3)

Inpatients aged ≥70 years had experienced a relatively high mortality rate during the study period. Among this group, in-hospital deaths were similar between the NFV (14/83 (16.8%)) and FV (10/53 (18.86%)) groups (OR, 1.14; 95% CI (0.46–2.80), *p* = 0.81).

About 77 fully vaccinated patients, 64 (83.1%) were vaccinated with m-RNA vaccines, and 13 (16.9%) with Adeno-vector vaccines with eleven patients death in both groups. There were 10/64 (15.6%) patients death in m-RNA group in front of 1/13 (7.7%) in the Adeno-Vector group (*p* = 0.67). 

### 3.7. One-Month Mortality

The one-month mortality assessment revealed that 22 (5.3%) patients in the NFV group died, compared with 11 (14.3%) in the FV group (Table 5). However, the preliminary analysis also showed a higher age and number of comorbidities in the FV group compared to the NFV group (Table 2). Likewise, Table 5 shows the percentage of the progressive increase in mortality among the unvaccinated in the highest age groups. A multivariate Cox analysis of mortality at 30 days of admission was performed to assess the relative risk of mortality during the first 30 days of admission; this was adjusted for the following variables presented at admission: only increasing age (hazard ratio (HZ), 1.1 (1.06, 1.14)) and the presence of haematological disease at admission (HZ, 2.99(1.26, 7.11)) were significant independent risk factors for higher mortality during the first 30 days of hospital income; vaccination was not significant once adjusted for other covariates.

### 3.8. Length of Hospital Stay

A benefit of vaccination was observed in the Cox multivariate analysis of the subgroup of 440 patients who did not die during the first 30 days of hospital stay. After adjusting for other covariates, the variables significantly related to the probability of discharge were age (older to younger age: HZ, 0.98 (0.97, 0.99)) and vaccination status (Table 6), with the probability of discharge being 42% higher in the vaccinated compared to the unvaccinated group (HZ, 1.42 (1.05, 1.91)). The median hospital stay was 5 (4, 6) days for the vaccinated versus 6 (5, 7) days for the unvaccinated group (Figure 2).

## 4. Discussion

In this population study, patients that required hospital admission because of SARS-CoV-2 infections were older fully vaccinated patients, and younger unvaccinated patients that suffered more severe infections with higher inflammatory tests.

Fully vaccinated patients under 60 years of age did not require hospital admission. The mortality rate at 30 days of admission was lower and the median length of stay was shorter in the FV group compared to those in the NFV. Only older age and the presence of haematological disease at admission were independent risk factors for higher mortality rates during the first 30 days of hospital income. Although FV patients had a higher mortality index, vaccination was not significant once adjusted for other covariates.

The number of new COVID-19 hospital admissions in HUF (which currently serves a population of more than 200,000) decreased from March to October 2021. This is because the Spanish vaccination program had started several weeks before, starting with older adults. This could explain why the majority of hospital admissions comprised of unvaccinated younger people; the few FV patients who were admitted were older with several comorbidities. The article by Havers et al. [13] showed a similar trend in the U.S. population wherein NFV individuals had a 17-fold risk of hospitalisation compared with the FV individuals (including data from the summer of 2021, when the Delta variant was the dominant strain); the trend in Spain during the study period was similar.

The majority of COVID-19 patients hospitalised in the HUF were considered NFV (84.6%), had a mean age of 21 years less, and had fewer comorbidities than the FV group. Despite these facts, at admission, the proportion of bilateral pneumonia, the rate of severe COVID-19, and oxygen requirements were higher among the NFV patients, in whom an increased inflammatory state was also observed. These characteristics have also been described by Olivier et al. [14].

As aforementioned, blood inflammatory parameters tend to be higher in the NFV group, even though significant differences were only observed in the ferritin levels. In this regard, both serum iron and transferrin are found to be inversely correlated with proinflammatory cytokines such as interleukin-6; thus, low serum iron and transferrin levels are associated with a poor prognosis in SARS-CoV-2 infection cases [15]. Moreover, elevated serum ferritin levels have also been linked to an increased risk of severe COVID-19 [16], ICU admission, and mortality rate [17,18].

In our study, the total number of deaths in the Fuenlabrada city population, both during hospital income and at one month after discharge, was higher in the FV group. This trend is interesting, although it was not found in the multivariate analysis results. This shows that vaccination is a proxy for risk patients who have been prioritised during the vaccination campaign. Therefore, the fact that vaccination is able to partially or completely offset the impact of comorbidities confirms its usefulness. This trend has also been described in other studies [14]. Similar findings have also been observed by other authors such as Brosh–Nissimov et al. [19], who retrospectively analysed a cohort of 152 FV COVID-19 patients from 17 Israeli hospitals, admitted at least seven days after the second dose of the BNT162b2 mRNA vaccine. A total of 34 (22%) patients finally died (compared to an in-hospital mortality of 10 [18.9%] in the FV group in our study), concluding that severe and fatal COVID-19 occurred only in a relatively lower proportion of FV patients, in spite of the high rates of comorbidities and immunosuppression in their cohort.

Age is one of the most important factors associated with death in patients with COVID-19. Complete vaccination prevents death, and fewer fully vaccinated patients needed hospital admission. However, among these fully vaccinated patients, SARS-CoV-2 infection was related to a high mortality rate, which, in turn, was likely related to the older age and higher number of comorbidities in this group. This idea is congruent with those presented in other reports. Muthukrishnan et al. [20] found that in a hospital-based cross-sectional study on vaccination status and COVID-19-related mortality, there was a significantly lower COVID-19 mortality rate among FV patients who had received the ChAdOx1 nCoV-19/AZD1222 vaccine, although the authors noted greater mortality among those aged >65 years. This increase was in spite of their vaccination status.

The main limitation of our study was its retrospective design. Furthermore, it was conducted before the Omicron variant of SARS-CoV-2 became widespread. During the study period, the Delta variant was the most prevalent strain in Madrid. Another limitation was that the anti-spike antibody titres were not measured in the vaccinated patients. Perhaps severe infections in the vaccinated patients did not have a positive serology.

In addition, it was clear that among patients hospitalised because of COVID-19, complete vaccination was associated with less severe disease forms (even in the presence of comorbidities), an earlier discharge date, and lower mortality rates during the first 30 days of hospital stay. Fully vaccinated patients under 60 years of age who needed hospital admission because of COVID-19 were considered as exceptions.

Older age and the presence of haematological disease was related to a higher mortality rate during the first 30 days of hospitalisation.

COVID-19 vaccination has reduced hospital income and deaths, but hospital admissions are still necessary in not vaccinated patients and in the vaccinated older group of patients. Because patients with the same age, same vaccine status, and same risk factors have a different evolution (not hospitalised, hospitalised, and deaths), the reasons are still unknown.

## Figures and Tables

**Figure 1 viruses-14-02284-f001:**
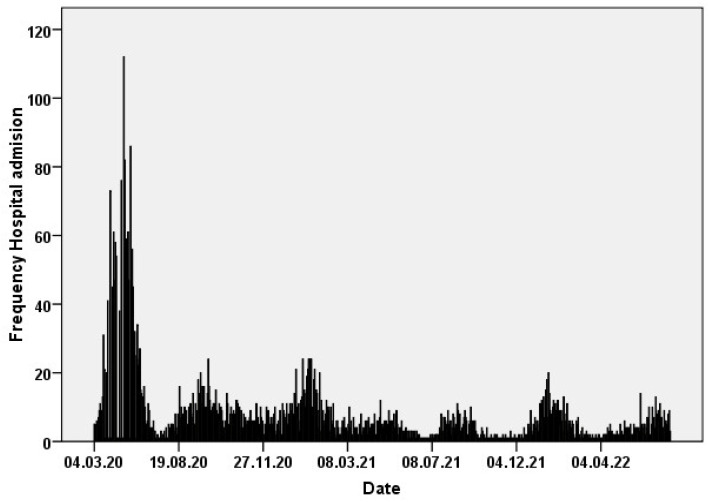
COVID-19 pandemic and hospital admissions in the area of Fuenlabrada.

**Figure 2 viruses-14-02284-f002:**
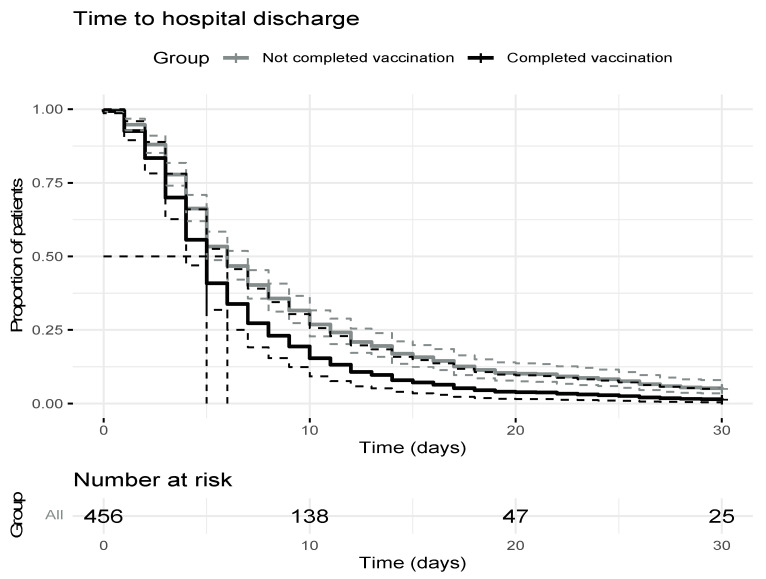
Time to hospital discharge.

**Table 1 viruses-14-02284-t001:** Hospital admissions, mean age of patients, ICU admissions, and deaths in hospital due to COVID-19 patients in HUF along the seven pandemic waves.

Waves	Hospital Admissions	Mean Age, (SD) Years	ICU (%)	Deaths in Hospital (%)	Deaths in Patients >70 Years (%)	Date
First	1586	63 (14.9)	85 (5.4)	154/1570 (9.8)	108/499 (21)	4 March 2020 to 14 July 2020
Second	915	60 (17.6)	76 (8.3)	77/910 (8.5)	46/253 (18.1)	15 July 2020 to 25 November 2020
Third	853	65 (16.6)	75 (8.8)	90/833 (10.8)	62/305 (20.3)	26 November 2020 to 28 February 2021
**Fourth**	**306**	**61 (14.6)**	**44 (14.7)**	**16/306 (5.2)**	**14/85 (16.4)**	**1 March 2021 to 30 June 2021**
**Fifth**	**194**	**56 (20.1)**	**23 (11.5)**	**11/194 (5.6)**	**10/51 (19.6)**	**1 July 2021 to 27 September 2021**
Sixth	538	66.5 (17.4)	34 (6.3)	34/538 (6.3)	23/191 (12)	1 October 2021 to 4 April 2022
Seventh	344	74.5 (14.8)	5 (2)	16/344 (4.3)	14/166 (8.4)	6 April 2022 to 4 July 2022
Total	4736					

Values are no. (%) except as indicated. ICU: intensive care unit; COVID-19: coronavirus disease; HUF: Hospital Universitario de Fuenlabrada; SD: standard deviation. Forth and Fifth Waves are the periods of study.

**Table 2 viruses-14-02284-t002:** Demographic and comorbidities related to the not vaccinated and vaccinated cohorts *.

Variable		No.of Patients	Not Vaccinated		Vaccinated		*p* Value
**Vaccination**		500	423	84.6%	77	15.4%	<0.01
Age (years)		500	423	55 (43–67)	77	76 (67–87)	<0.01
Age	Under 40 years	75	75/423	17.7%	0/77	0%	<0.01
	41–50	94	93/423	22%	1/77	1.3%	
	51–60	94	88/423	20.8%	6/77	7.8%	
	61–70	101	84/423	19.9%	17/77	22.1%	
	71–80	79	60/423	14.4%	19/77	24.7%	
	Over 80	57	23/423	5.4%	34/77	44.2%	
Sex (M/F)		500 (272/228)	423 (236/137)	55.8%/44.2%	77 (36/41)	46.8%/53.2%	0.17
BMI (kg/m^2^)		368	328	29.3 (26.6–33.2)	60	28.8 (25.4–33.1)	0.26
Charlson scale		476	400	0 (0–1)	76	2 (0.25–3)	<0.01
Hypertension		211/497	154/420	36.7%	57/77	74%	<0.01
Diabetes		88/498	56/421	13.3%	32/77	41.6%	<0.01
Ischemic cardiopathy		26/497	13/421	3.1%	13/76	15.3%	<0.01
Heart failure		27/496	17/419	4.1%	10/77	13%	<0.01
Smoker		29/500	24/423	5.7%	5/77	6.5%	0.96
COPD		41/500	32/423	7.6%	9/77	11.7%	0.25
Haematological disease		30/495	22/418	5.2%	9/77	11.6%	0.20
Oncological disease		28/492	20/415	4.8%	8/77	10%	0.23
Autoimmune disease		16/498	15/421	3.6%	1/77	1.3%	0.48
HIV		1/492	1/460	0.2%	0/77	0%	0.66
Dementia		24/496	9/419	2.1%	15/77	19.5%	<0.01
Nursing home		17/486	9/413	2.2%	8/77	15%	<0.01

* Values are no. (%) except as indicated.; M: male; F: female; BMI: body mass index; COPD: chronic obstructive pulmonary disease; HIV: human immunodeficiency virus; ICU: intensive care unit; VM: venturi mask; NRB: non-rebreather mask.

**Table 3 viruses-14-02284-t003:** Clinical characteristics in the not vaccinated and vaccinated cohorts *.

Variable	No. of Patients	Not Vaccinated		Vaccinated		*p* Value
Initial chest radiography						
No pneumonia	81/451	51/376	13.6%	30/75	40%	<0.01
Unilateral pneumonia	79/451	66/376	17.6%	13/75	17.5%	
Bilateral pneumonia	291/451	259/376	68.9%	32/75	42.7%	
Severity						
Mild (without pneumonia)	69/490	43/415	10.4%	26/75	34.7%	<0.01
Moderate (pneumonia without severe symptoms)	184/490	163/415	39.3%	21/75	26%	
Severe (Tachipnea, saturation under 93% or more than 50% affected lungs)	237/490	209/415	50.4%	28/75	37.3%	
Highest oxygen requirements						
Not oxygen	77/492	66/416	15.9%	11/76	14.3%	0.23
Low flow oxygen	260/492	216/416	51.9%	44/76	57.9%	
VM 28%	2/492	1/416	0.2%	1/76	1.3%	
VM 31%	5/492	5/416	1.2%	0	0%	
VM 35%	34/492	26/416	6.3%	8/76	10.5%	
VM 50%	91/492	83/416	20%	8/78	10.5%	
NRBM 100%	23/492	19/416	4.6%	4/76	5.3%	
Analytical findings						
Highest CRP (mg/mL)	490	415	9 (3.8–13)	75	10.2 (3.8–17.2)	0.38
Highest IL6 (pg/mL) in hospital	372	317	38.4 (12–104)	55	40 (6.7–65)	0.88
Highest IL6 (pg/mL) pretreatment with tocilizumab	271	229	26.4 (8.4–59.5)	42	14.9 (5.3–54.6)	0.20
Lowest lymphocyte (value/mcL)	492	416	875 (650–1210)	76	930 (572–1320)	0.43
DD (mg/dL)	486	411	925 (565–1625)	75	879 (437–2287)	0.71
LDH (mg/dL)	487	412	323 (257–412)	75	313 (229–436)	0.83
Highest ferritin (ng/dL)	460	391	700 (336–1268)	69	351 (144–739)	<0.01

* Values are expressed as no. except as indicated. * CRP: C-reactive protein; DD: D-dimer; LDH: lactate dehydrogenase; IQR: range interquartile; IL6: interleukin 6.

**Table 4 viruses-14-02284-t004:** Pharmacological treatments received by the both not vaccinated and vaccinated cohorts *.

Treatments	Total Patients	Not Vaccinated (%)	Vaccinated (%)	*p* Value
Remdesivir	7/500	5/423 (1.2)	2/77 (2.6)	0.29
Tocilizumab	228/500	202/423 (47.8)	26/77 (33.8)	0.025
Imatinib	15/500	15/423 (3.5)	0/77 (0.0)	0.093
Baricitinib	59/500	49/423 (11.6)	10/77 (10.0)	0.72
Anakinra	5/500	5/423 (1.18)	0/77 (0.0)	1
LMWH	478/500	404/423 (95.5)	74/77 (96.1)	1
Corticosteroids	450/500	384/423 (90.8)	66/77 (85.7)	0.17

* Values are no. (%) except as indicated. LMWH: low-molecular-weight heparin.

**Table 5 viruses-14-02284-t005:** Hospital stay and evolution in the not vaccinated and vaccinated cohorts *.

Variable	No. of Patients	Not Vaccinated		Vaccinated		*p* Value
Hospital stay (days)	483	407	6 (4–11)	76	6 (4–12)	0.63
ICU stay (days)	67	63	17.8 (7–26)	4	17 (4–18)	0.26
ICU admission	67/500	63/423	14.9%	4/77	6%	0.018
Under 40 y	12/75	12/75	16%	0/0	0%	
41–50 y	12/94	12/93	12.9%	0/1	0%	
51–60 y	9/94	8/88	9%	1/6	16.6%	
61–70 y	18/101	18/84	21.4%	0/17	0%	
71–80 y	14/79	12/60	20%	2/19	10.5%	
Over 80 y	2/57	1/23	4.3%	1/34	2.9%	
Death in hospital	27/500	17/423	4%	10/77	13%	<0.01
Under 40 y	0/75	0/75	0%	0/0	0%	<0.01
41–50	0/94	0/93	0%	0/1	0%	
51–60	0/94	0/88	0%	0/6	0%	
61–70	3/101	3/84	3.5%	0/17	0%	
71–80	11/79	8/60	13.3%	3/19	15.7%	
Over 80 years	13/57	6/23	26%	7/34	20.5%	
Death one month after discharge	33/500	22/418	5.3%	11/77	14.3%	<0.01
Under 40 y	0/75	0/75	0%	0/0	0%	
41–50 y	0/94	0/93	0%	0/1	0%	
51–60 y	1/94	1/88	1.1%	0/6	0%	
61–70 y	4/101	3/84	3.5%	1/17	5.8%	
71–80 y	13/79	10/60	16.6%	3/19	15.7%	
Over 80 y	15/57	8/23	34.7%	7/34	20.5%	

* Values are no. (%) except as indicated. d: days; y: years; ICU: intensive care unit.

**Table 6 viruses-14-02284-t006:** Cox analysis: risk of death in the first 30 days of admission and risk of being discharged alive in the first 30 days of admission.

Risk of Death in the First 30 Days of Admission	Univariant Analysis				Multivariant Analysis			
	HR	IC 2.5%	IC 97.5%	*p* Value	HR	IC 2.5%	IC 97.5%	*p* Value
Completed Vaccination (ref = no)	3.38	1.57	7.30	<0.01				
Age (years)	1.10	1.06	1.14	<0.01	1.10	1.06	1.14	<0.01
Charlson (ref = ≤ 1)	4.51	2.11	9.65	<0.01				
Hypertension (ref = no)	2.07	0.93	4.61	0.08				
Heart failure (ref = no)	3.11	1.17	8.23	<0.03				
Haematological disease (ref = no)	3.46	1.46	8.23	<0.01	2.99	1.26	7.11	<0.02
Dementia	5.45	2.29	12.95	<0.01				
Risk of being discharged alive in the first 30 days of admission	Univariant analysis				Multivariant analysis			
	HR	IC 2.5%	IC 97.5%	*p* Valor	HR	IC 2.5%	IC 97.5%	*p* Value
Completed Vaccination (ref = no)	0.98	0.76	1.28	0.90	1.42	1.05	1.91	<0.025
Age(years)	0.985	0.98	0.99	<0.01	0.98	0.97	0.99	<0.01
Hypertension (ref = no)	0.80	0.66	0.97	0.03				
Haematological disease (ref = no)	0.64	0.42	1.0	0.051				

## Data Availability

Database belongs to the Hospital Universitario de Fuenlabrada. Restrictions apply to the availability of these data. Data were obtained from patients hospitalised and are available [josemanuel.ruiz@salud.madrid.org] with the permission of Gerencia Hospital Universitario de Fuenlabrada.

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
