# Peer review of "Comparative Study of Vaccinated and Unvaccinated Hospitalised Patients: A Retrospective Population Study of 500 Hospitalised Patients with SARS-CoV-2 Infection in a Spanish Population of 220,000 Inhabitants"

_viruses, 2022, doi:10.3390/v14102284_

Round 1

Reviewer 1 Report

This work is one of the few in which a repeated truth is being quantified, and that is that vaccines against COVID-19 have protected people from dying. In addition, a detailed description of the clinical parameters of the patients, laboratory tests, and other important ones are shown, allowing comparisons between the groups of patients included in the study. A complete statistical analysis of the data is also carried out, which gives reliability to the study, which must not have been easy to conceive at the height of the pandemic. Thank you for publishing this information. Authors need to refine their English and watch out for words like "hospitalised" repeated often in the text.

Author Response

Please , see the attachment

Reviewer 2 Report

Interesting information, good statistical analysis.

Remarks: 

section 3.6. Second paragraph. "At the time of hospital discharge, COVID-19 severity was assessed. We found that severe COVID-19 was more frequent among the NFV patients (209 [50.4%] than among FV 28 [37.3%]; OR, 1.70; 95% CI [1.02–2.82]; p = 0.001) (Table 5)"

Invalid table reference. The data from Table 3 are presented. 

section 3.7. Now: "The one-month mortality assessment revealed that 22 (5.3%) patients in the NFV group died, compared with 11 (14.3%) in the FV group. However, the preliminary analysis also showed a higher age and number of comorbidities in the FV group compared to the NFV group (Table 5)."

 It is difficult to find confirmation in different tables. I think it's better to write like this: "The one-month mortality assessment revealed that 22 (5.3%) patients in the NFV group died, compared with 11 (14.3%) in the FV group (Table 5). However, the preliminary analysis also showed a higher age and number of comorbidities in the FV group compared to the NFV group (Table 2)".

This will improve the article:

If the authors have data on a specific vaccine (BNT162b2, m-RNA-1273, AZD1222, Ad26.COV2.S) in each patient, you need to add "name of vaccine" or "type of vaccine (m-RNA or Adeno-vector)" to the table 5 in "Death in hospital - Vaccinated" and "Death one month after discharge - Vaccinated", to the table 6 with Cox analysis. It is important to know whether the risk of death depends on the name or type of vaccine. 
